# A Novel Method of Fabricating Al-V Intermetallic Alloy through Electrode Heating

**Heli Wan** [1,2,3,4], **Baoqiang Xu** [1,2,3,*], **Lanjie Li** [4], **Bin Yang** [1,2,3], **Dongming Li** [4] **and Yongnian Dai** [1,2,3]

1   National Engineering Laboratory for Vacuum Metallurgy, Kunming University of Science and Technology, Kunming 650093, China; wanheli@163.com (H.W.); kgyb2005@126.com (B.Y.); xiyuwangli@126.com (Y.D.)
2   Research Center of Engineering on Aluminum Industry of Yunnan Province, Kunming 650093, China
3   Faculty of Metallurgical and Energy Engineering, Kunming University of Science and Technology, Kunming 650093, China
4   Cheng de Iron and Steel Company, Chengde 067002, China; lilanjie20040014@163.com (L.L.); wfk15587221776@126.com (D.L.)
*   Correspondence: kmxbq@126.com; Tel.: +86-13608864121

**Abstract:** A novel method was developed to produce AlV55 alloy through reducing impurities content and component segregation with electrode assisted heating technology. This new process synergistically integrates a few low-cost process techniques, including granulation, mixing, and electro-heating to produce AlV55 alloy. During the heating process, the CaO is used as an additive in the raw materials. The uniform of AlV55 alloy composition and low impurities content are effectively controlled by this process. The analysis results show that Si (0.13 wt%), Fe (0.22 wt%), N (0.007 wt%), C (0.078 wt%), and O (0.051 wt%) impurities in the AlV55 products were reduced, which met the commercial standard (TS/T 579-2014), and V content ranged from 57.5 to 58.5 wt% when the Al/$V_2O_5$ mass ratio was 0.94:1. This method can realize the controllability of the reaction process and is suitable for large-scale industrial production.

**Keywords:** Al-V alloy; vanadium pentoxide; aluminothermic reduction; electrode heating

## 1. Introduction

Vanadium and aluminum vanadium (AlV) alloys are of the most important master alloys, which are promising materials in high temperature alloy and some special alloys fields. The addition of vanadium and aluminum can significantly improve the performance of heat resistance and cold working on the titanium alloy and achieves high mechanical strength [1,2]. Therefore, AlV alloy and titanium alloy are widely used in the military and the aerospace industries [3,4]. The high purity and the uniform intermediate alloys are of critical importance for fabricating titanium alloy with excellent performance [5,6], especially for the aerospace and aviation industry [7–9]. The uniform composition and the minimum impurities content are conducive for using AlV as an intermediate alloy, and they are desired to fabricate titanium alloy [10].

Over past decades, the AlV alloy was mainly produced by the aluminothermic reaction with vanadium oxide ($V_2O_5$ or $V_2O_3$) and aluminum [11]. At present, the production methods of the AlV alloy include two steps process, aluminothermic reduction and SHS-metallurgy [12]. The two steps process is namely the aluminothermic-vacuum melting method [13,14]. The first step is to prepare high-vanadium aluminum alloy through aluminothermic reduction; the product contains 85 wt% vanadium (AlV85 alloy). The second step is to melt the AlV85 alloy with a certain proportion of aluminum addition. Finally, the AlV55 alloy product is obtained, which has the advantages of uniform

composition and low impurity content. However, the production process is complex, and equipment requirement is rigorous. Aluminothermic reduction is a conventional method, and its raw materials are the mixture of $V_2O_5$, Al powder, and $CaF_2$ [15]. The method is simple, however, which has critically influenced the composition and the oxygen content of alloy products. The method of SHS-metallurgy is the use of $V_2O_5$ and Al powder as raw materials. This method directly produces AlV alloys in vacuum. The reaction process of controllability is weak, and the uniformity of the alloy composition is undesired.

Among all these processes, the most universal process for AlV55 alloy production is aluminothermic reaction using $V_2O_5$ and Al powder as raw materials. The AlV55 alloy is produced by SHS-metallurgy, and the Al is the main component in vanadium-based alloy [16]. The component segregation of Al has great influence on alloy composition and further aggravates the malleability of the AlV55 alloy. Thus, it is important to elucidate the effects of composition alloys on the preparation of the AlV55 alloy process. The electrode-assisted heating technique to produce the AlV55 alloy is extremely important, however, which remains largely unexplored.

In this paper, a new process is proposed to prepare AlV55 alloy based on the electrode assisted heating process, which can effectively control heat release rate and improve the recovery of V. The two step feed is conducted in the process of preparation of the AlV55 alloy during the assisted heating process. The concentrated release of heat is effectively avoided, and the V loss is reduced. In addition, the furnace slag temperature can be adjusted by adding a heat source from electrode-assisted heating. Thus, the slag and the alloy are sufficiently separated, and the content of V can be reduced to below 0.1% in the slag. Furthermore, this method can realize the controllability of the reaction process and is suitable for large-scale industrial production.

## 2. Experimental Details

### 2.1. Materials

$V_2O_5$ granules were obtained from Chengsteel Group Co. Ltd. (Chengde, China), and the commercial grade Al granules and CaO powder were used in the experiment. $Al_2O_3$ was mainly used for lining the furnace, and the slag ($CaO \cdot Al_2O_3$) was used as a lining material in the top part of furnace body, where it could not come into contact with the materials. The AlV55 particles came from products whose particle sizes did not meet commercial application standards. The form and the purity of raw materials are shown in Table 1.

**Table 1.** The raw materials in the experiments.

| Material | Form | Purity or Content /% [a] | Particle Size |
|---|---|---|---|
| $V_2O_5$ | Granules (technical grade) | $V_2O_5 > 99\%$; Fe < 0.05%; Si < 0.08% | 1–3 mm |
| Al | Granules (technical grade) | Al > 99.85%; Fe < 0.1%; Si < 0.08% | 1–3 mm |
| $Al_2O_3$ | Lump (technical grade) | $Al_2O_3 > 99\%$; Fe < 0.05%; Si < 0.08% | 2–5 mm |
| CaO | Powder (technical grade) | CaO > 98.5%; Fe < 0.1%; Si < 0.4% | ≤50 μm |
| AlV55 | Granules | V: 57–59%; Fe < 0.20%; Si < 0.20%; C < 0.1%; N < 0.04%; O < 0.18%; Al: remainder | ≤1 mm |

[a] Concentration of solution.

### 2.2. Experimental Procedures

In the experiment, V and $Al_2O_3$ were formed through the self-propagating reaction between $V_2O_5$ and Al granules. The released heat during the reduction process was enough to melt the excessive Al granules in the furnace, which then formed the AlV55 alloy product. In order to separate the slag and the AlV55 alloy, electrode heating was employed to provide the heat, since this reduced the solidification rate of the AlV55 alloy.

The schematic of the experimental procedures is presented in Figure 1. $V_2O_5$ granules and Al granules with different mass ratios were mixed in the high efficient sealed blender for 10 to 30 min, and CaO was used as an additive. The mixed materials were ignited by the electrode in the arc furnace. During the raw materials reaction process, the materials were continuously added to the furnace. In the meantime, the heat was provided by electrode heating for 5~20 min. The AlV55 alloy was separated completely from the slag when the product was cooled for 72 h. Commercial grade AlV55 alloy was obtained, and the slag was used as the refractory for the furnace lining.

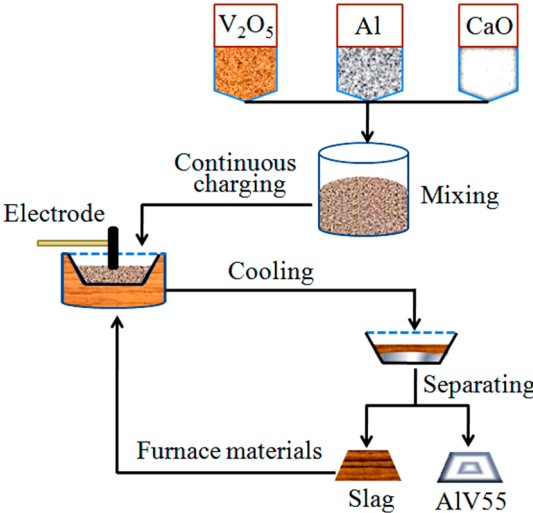

**Figure 1.** Schematic diagram of the experiments.

As mentioned before, Al would be ignited and volatized during the electrode heating process. In order to prevent the segregation, the AlV55 alloy with Al consuming the different mass ratios of raw materials were investigated in this work and are listed in Table 2.

**Table 2.** The composition of raw materials.

| No. | $M_{V2O5}:M_{CaO}$ (kg) | $M_{Al}:M_{V2O5}$ | Heating Time (min) |
|-----|-------------------------|-------------------|--------------------|
| 1 | 2.5:1 | 1:1 | 10 |
| 2 | 2.5:1 | 0.96:1 | 10 |
| 3 | 2.5:1 | 0.94:1 | 10 |
| 4 | 2.5:1 | 0.92:1 | 10 |
| 5 | 2.5:1 | 0.9:1 | 10 |

*2.3. Characterization*

The concentrations of oxygen/carbon/nitrogen (O, C, N) and ferrum/silicon (Fe, Si) elements in the AlV55 products were measured by LECO TCH 600/LECO C/S 230 and inductively coupled plasma atomic emission spectrometer (ICP-AES) with a PS-6 PLASMA SPECTROVAC, BAIRD (Milwaukee, WI, USA), respectively. The phases of products were identified by XRD using a Rigaku (Tokyo, Japan) D/max-3B X-ray diffractometer equipped with graphite-monochromatized Cu Ka radiation. SEM of S3400 N equipped with an energy dispersive spectrometer (EDS) were used to detect the element distribution of Al and V by the spot-scan method.

**3. Results and Discussion**

*3.1. Theoretical Analysis*

The exothermal intensity is described by the unit thermal effect ($q$), and the reaction can be finished when "$q$" reaches a certain numerical "$q^\theta$". According to the trim "hot autumn day" rule [17], the

condition of heat extraction is: $|q| \geq q^{\theta} = 2300$ J/g [18]. In this work, the approximate heat of the reaction was calculated by the volatilization of the materials in the reaction process, and the main exothermic reaction is provided as follows:

$$2V_2O_5(s) + 10Al(s) = 4V(s) + 5Al_2O_3(s); \ \Delta H_{298}^{\theta} = -491.8 \text{ KJ/mol} \tag{1}$$

The experience indicates that the Al thermal reaction occurred at about 943 K according to the Kirchhoff's formula [19]. As a result, $|q| = 4988$ J/g $> q^{\theta}$. As was expected, the aluminothermic reaction was very intense during the preparation of Al-V alloys. A significant amount of heat was released instantly, which resulted in the volatilization of Al. Under this condition, about 5% of the raw materials ($V_2O_5$, Al) were volatilized due to the high temperature, and therefore a certain amount of CaO and AlV55 alloy powder was added to reduce the effect of boiling on the volatilization of the raw materials. Finally, the volatile amount of raw materials was less than 1%, which was almost negligible. However, the instant release of the aluminothermic reaction resulted in an inhomogeneous composition, because the remaining heat was insufficient. Additionally, cracks also appeared.

Generally, the inhomogeneous composition of the AlV55 alloy is caused by the inhomogeneous grain size, and the Al-V state diagram is shown in Figure 2. In the experiment, 35% of Al was dissolved in the solid V, and the solubility of V in solid Al was 0.27% at 923 K [20]. The intermediate compounds of the AlV alloy included $AlV_3$, $Al_3V$, and $Al_8V_5$, and all the intermediate compounds of the alloy were decomposed when they were melting. Because the crystal package of the AlV55 alloy was formed at 2083 K, the phase of V was generated in the liquid phase. However, when the temperature dropped to 1943 K, the peritectic reaction was as follows:

$$\text{Liquid} + V \rightarrow Al_8V_5$$

$Al_8V_5$ is a larger crystal particle compared to $AV_3$, and in the experiment, $Al_3V$ was formed in the primary phase [21,22]. This caused the segregation of the AlV55 alloy composition segregation in the peritectic zone during the preparation process. Therefore, the cracks and holes appeared in the alloy during the solidification process. Impurities such as oxygen and nitrogen diffused through the cracks into the alloy. Thus, in order to reduce the impurities, this experiment adopted the electrode heating process, which could also improve the system temperature gradient and the segregation of the AlV alloy.

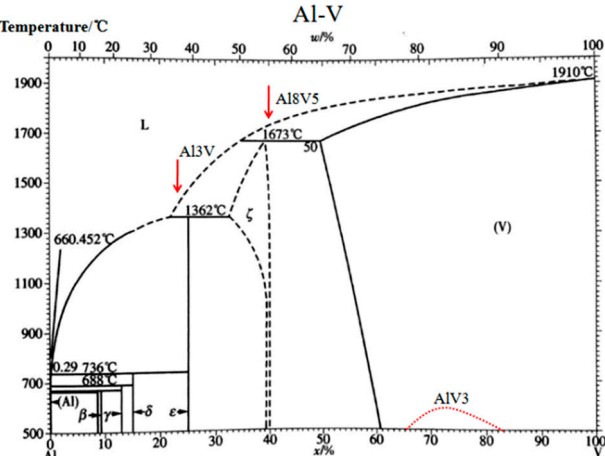

**Figure 2.** Phase diagram of the Al-V system.

The electrode heating method was employed to control the system temperature in order to reduce the large crystal particles of $Al_8V_5$. The reduction can be represented by the following reactions (2)–(5):

$$V_2O_5 + 2Al = VO_2 + Al_2O_3 \tag{2}$$

$$6VO_2 + 2Al = 3V_2O_3 + Al_2O_3 \tag{3}$$

$$V_2O_3 + Al = VO + Al_2O_3 \tag{4}$$

$$VO + Al = V + Al_2O_3 \tag{5}$$

HSC6.0 (Thermodynamic simulation software) was used to calculate the thermodynamics of the above equations [23]. The Gibbs free energy changes with the temperatures of the above reactions (2)–(5) were calculated, and the reaction curves are shown in Figure 3. The $\Delta G$ indicates that the reaction of $VO_2$, $V_2O$, and VO could occur between 273 K and 2073 K, and VO was the most difficult to reduce, which also illustrates that reaction (5) could easily occur compared to other reactions. This implies that increasingly more $VO_2$ would be formed. It has been reported that Al could reduce $VO_2$ at the temperatures above 1047 K, and $VO_2$ could be further gradually reduced to VO in molten slag [24]. VO could also be further reduced by Al to form the AlV alloy.

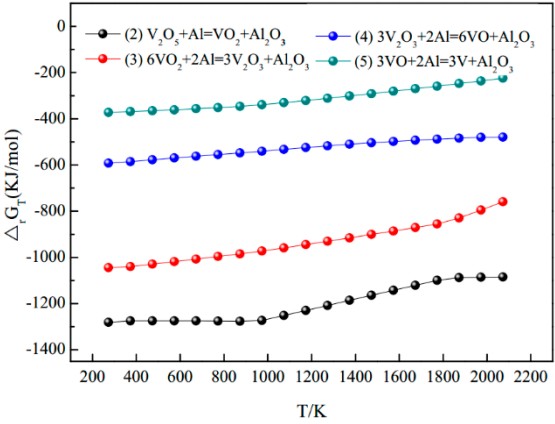

**Figure 3.** The $\Delta_r G_T$-$T$ curves of reactions (2)–(5) in standard conditions.

Additionally, it should be noted that a linear thermal reaction was observed for the $CaO$-$Al_2O_3$ system at 1273–1923 K [25], which suggests that $CaO$-$Al_2O_3$ was easily formed with the increase of temperature, and that the increase of temperature was beneficial for the reduction reaction. Consequently, we set the temperature range for the experiment from 1073 K to 2073 K.

*3.2. Effect of the Materials Composition*

CaO could not only reduce the melting point and the viscosity of slag but could also improve the fluidity of slag [26]. According to the phase diagram of the $CaO$-$Al_2O_3$ binary system, the melting point was about 1663 K [27]. The $(CaO)_x \cdot (Al_2O_3)_y$ ($CaO \cdot Al_2O_3$, $CaO \cdot 2Al_2O_3$) with a lower melting point could be formed by CaO and $Al_2O_3$ when CaO was added in the electrode heating process. In this work, the mass ratio of $V_2O_5$/CaO was fixed at 2.5:1. The surface tension of the alloy and the slag was separated during the self-propagating process, as shown in Figure 4a. Some of the AlV55 alloy existed on the surface of the slag, which caused the yield of the alloy to be reduced. The surface tension was notably improved, which allowed for the easy separation of the AlV55 and the slag, as shown in Figure 4b. Furthermore, the XRD pattern of the AlV55 alloy and the slag are presented in Figure 4c, and the experimental conditions are listed in Number 2 in Table 2. Aluminate was formed by CaO and $Al_2O_3$, and, in addition, the aluminothermic reaction was accelerated with the reduction of $Al_2O_3$ in the electrode heating process.

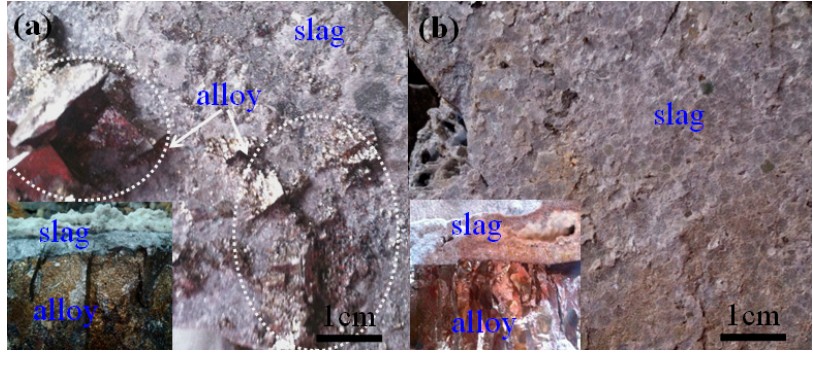

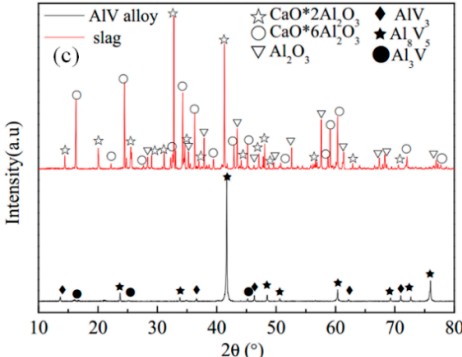

**Figure 4.** (**a**) Photograph of the alloy and the slag (heating time = 0 min); (**b**) photograph of the alloy and the slag (heating time = 10 min); (**c**) XRD patterns of the products obtained after reaction ($M_{Al}$:$M_{V2O5}$ = 0.96:1; heating time = 10 min).

SEM was employed to investigate the relationship between the initial materials composition and the final alloy microstructure, as presented in Figure 5. As exhibited in Figure 5a,b, irregular cracks or holes were observed in the samples prepared by this method. It should be noted that this typical composition segregation would only happen if there was excessive Al in the raw material, which could also lead to the generation of flaky and brittle $A1_8V_5$ crystals. Due to the increase of Al in the raw materials, it was proposed that the conversion process of Al-V compounds in the alloy was V–$AlV_3$–$A1_2V_3$–$A1_8V_5$ [28,29].

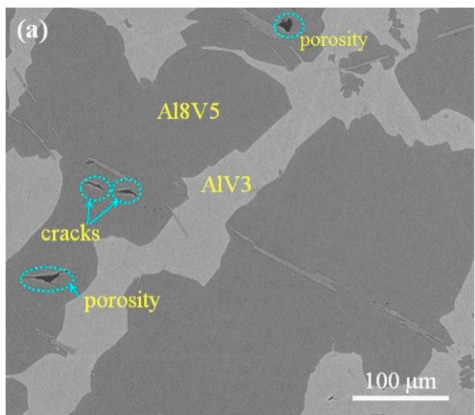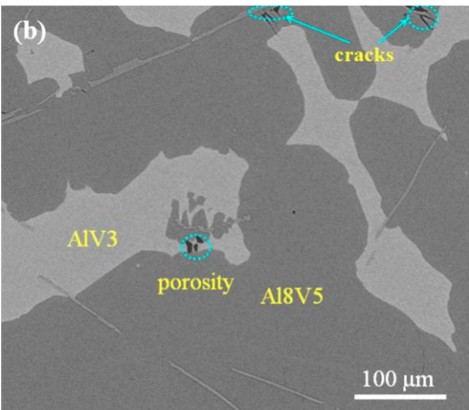

**Figure 5.** *Cont.*

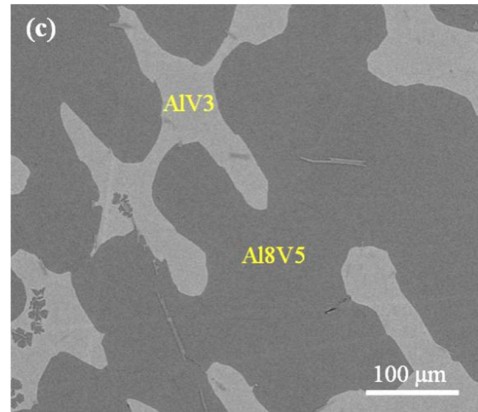
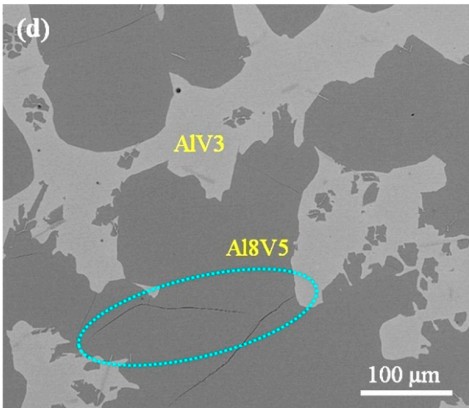
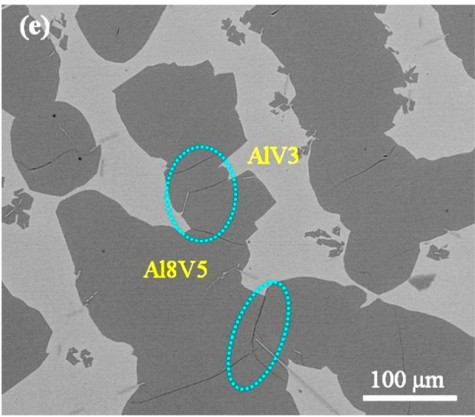

**Figure 5.** SEM images of AlV produced by different proportion of raw materials (**a**): $M_{Al}$:$M_{V2O5}$ = 0.9:1; (**b**): $M_{Al}$:$M_{V2O5}$ = 0.92:1; (**c**): $M_{Al}$:$M_{V2O5}$ = 0.94:1; (**d**): $M_{Al}$:$M_{V2O5}$ = 0.96:1; (**e**): $M_{Al}$:$M_{V2O5}$ = 1.0:1.

As shown in Figure 5c, the AlV55 alloy exhibited excellent performance when the mass ratio of Al/$V_2O_5$ was 0.94:1. The $AlV_3$ and the $Al_8V_5$ phase grew into a dendritic morphology in the AlV55 alloy. However, with decreasing Al, the alloy composition, inhomogeneous of the flaky $Al_8V_5$ crystal, was increased, and small cracks were formed gradually in the $Al_8V_5$ crystals regions. In addition, the small $Al_8V_5$ crystals entered the $AlV_3$ phase, as shown in Figure 5d,e. Therefore, in the experimental process, the mass ratio of Al/$V_2O_5$ was 0.94:1.

### 3.3. Effect of Electrical Heating

After a comprehensive evaluation of alternatives, the position of the electrode was fixed to a location above the slag layer during heating. The parameters of the graphite electrode were as follows: length (*L*) was 160 cm, diameter (*D*) was 25 cm, resistivity was 6.5–7.0 μΩ·m, density of the electrode was 1.6–1.7 g/cm$^3$, and carbon mass fraction was greater than 99.5%. The inner diameter of the bottom of the furnace was 100 cm (*D*1), the top was 150 cm (*D*2), the part below the slag surface was made of $Al_2O_3$, and the upper part was made of slag ($Al_2O_3$·CaO). The thickness was 30 cm, the diameter size of the base of the alloy ingot was about 40 cm ($d_2$), and the diameter at the top of the slag layer was about 80 cm ($d_2$). Figure 6 shows the assemblage used in the electrode heating process. By this method of heating, carbon impurity in the electrode could be prevented, and it was much easier for the separation of AlV55 product and slag. In addition, it was beneficial for the uniform composition and the V recovery. Heating was able to avoid rapid solidification, however, which resulted in more carbon diffuse into the alloy in long holding times. The temperature of the measurement results showed temperature range of the alloy was more than 1923 K (about 1980–2000 K) by the electrode heating, and it was mainly related with the heating time. However, the effects of electrode heating on temperature increase were not obvious, which were mainly to supplement the heat to improve the cooling rate of the alloy. Therefore, it was required to decide the right heating time in order to get purer AlV55 alloy.

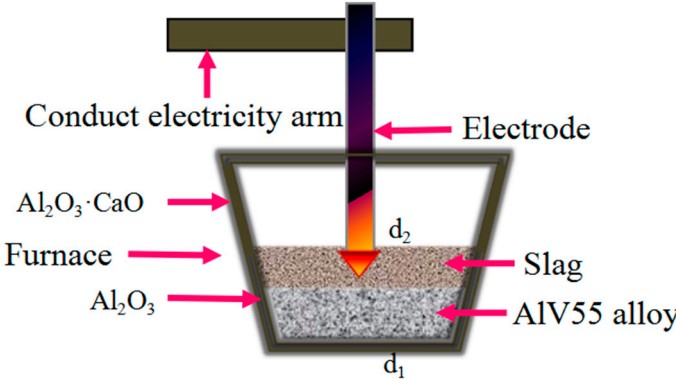

**Figure 6.** Assemblage used in the electrode heating process.

The main impurities in the product were Si, Fe, C, N, and O. The impurities concentrations obtained from different heating times are shown in Figure 7a. The impurities of N, O, and C could form $Al_2O_3$, AlN, and VC, respectively. Figure 7b indicates that the amount of V content in the alloy had no significant change when heating time ranged from 3–11 min. However, the content of C and V in the product was too high to meet the commercial standard (TS/T 579-2014) after heating for 15 min [30]. The requirements for AlV55 alloy for commercial application are shown in Table 3 [31].

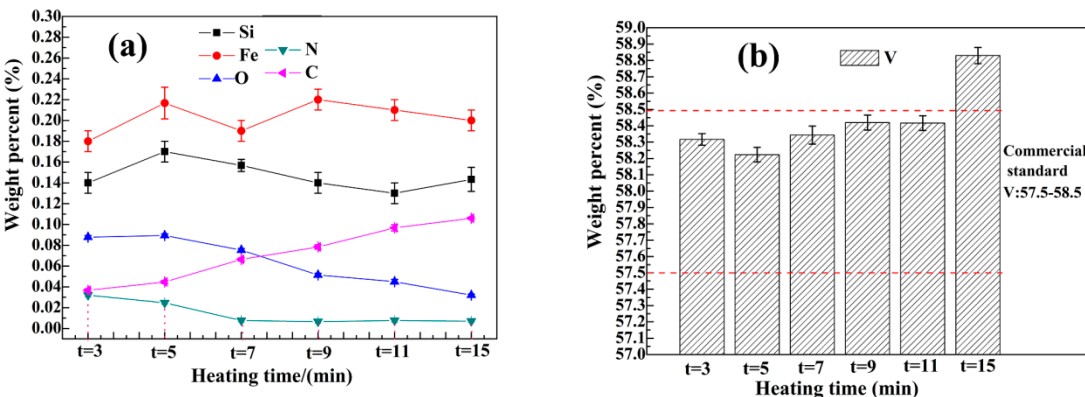

**Figure 7.** The composition of the product with different heating time (**a**): distribution of impurities in the sample; (**b**): V content in the sample.

**Table 3.** The composition of the AlV55 alloy for commercial application (TS/T 579-2014).

| Content of Element (%) | | | | | | |
|---|---|---|---|---|---|---|
| **V** | **O** | **C** | **Fe** | **Si** | **N** | **Al** |
| 57.5~58.5 | ≤0.18 | ≤0.1 | ≤0.25 | ≤0.25 | ≤0.04 | remainder |

According to Figure 7a, it is easy to notice that the heating time had more influence on O and C than it did on Si, Fe, and N in the AlV55 alloy. According to Figure 7a, the content of O was decreasing with the increasing heating time; this was because more and more vanadium oxide was reduced after longer heating. Meanwhile, the concentration of C increased when heating time extended. We attributed this to the impurity of C diffusion from the slag to the AlV55, which also indicates that the employment of electrode heating could not avoid the C diffusion completely. The concentration of V in AlV55 increased with the longer heating, as shown in Figure 7b. We attributed this phenomenon to two reasons. At first, V diffused into AlV55 from the slag by electrode heating. This proved that the content of V in the slag was decreased from 0.143 wt% to 0.009 wt% when the heating time was increased from 0 min to 15 min, as shown in Figure 8. Then, it was noted that some Al would volatilize during heating. Therefore, we concluded that the heating time should have been controlled at 7~9 min for the electrode heating process in order to obtain purer AlV55 product.

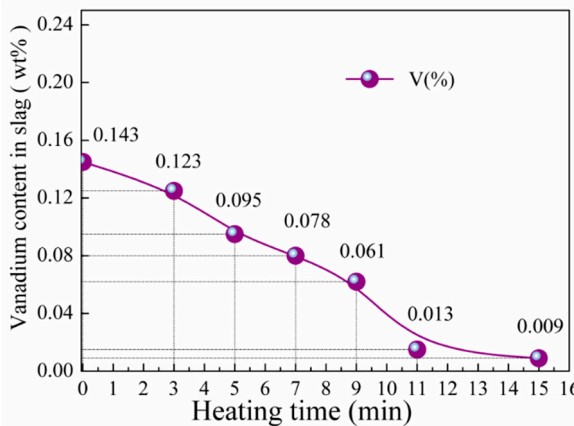

**Figure 8.** The content of V in slag with different electrode heating time.

Figure 9 shows the different microstructures of the AlV55 alloy after undergoing varying heating times in the electrode heating process. The $AlV_3$ phase (white) and the $Al_8V_5$ phase (gray) were embedded in each other. It can be seen that all the AlV55 alloys exhibited black pores or cracks when the heating time was 3 min or 5 min, as displayed in Figure 9a,b. The formation of pores or cracks was due to the generation of the brittle $Al_8V_5$ phase and the insufficient heat at the end of the reaction. As shown in Figure 9c, $Al_8V_5$ disappeared due to the reduced alloy cooling speed when heated for 9 min. As expected, the pores and cracks also disappeared, which resulted in the uniform distribution of $Al_8V_5$ and $A1V_3$ in the AlV55 alloy.

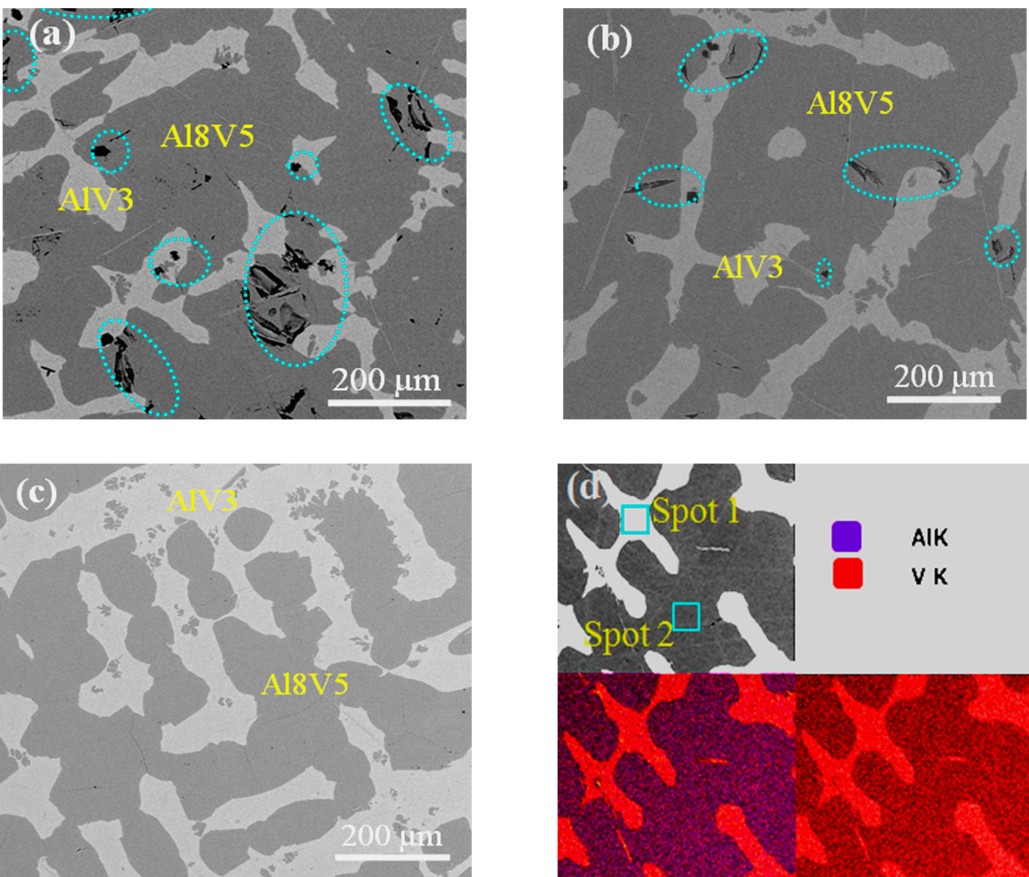

**Figure 9.** SEM micrographs of AlV55 alloy by different heating time (**a**): heating time = 3 min; (**b**): heating time = 5 min; (**c**): heating time = 7 min; (**d**): heating time = 9 min; and (**d**) the distribution of elements in the sample.

EDS was implemented to investigate the effect of the heating process on the uniformity of the AlV55 alloy composition. Figure 10 exhibits the distribution of Al and V in the product that was obtained after heating for 7 min. The EDS compositional analysis of the atomic distribution structure provided in Figure 9d reveals that the AlV55 alloy was a non-stoichiometric compound with V content. The purple is the distribution of Al (corresponding to the white region in Figure 9a–c). The mass ratios of Al and V in this area were 33 wt% and 67 wt% (point 1), respectively. The results indicate that the composition of the atoms was close to 1:3 (Al:V), and the alloy compounds in this region were mostly $AlV_3$. However, red represents the distribution of V (the gray areas in Figure 9a–c), and the mass ratios of Al and V were 46 wt% and 54 wt% (point 2), respectively. The results demonstrate that the composition of the atoms of Al:V were close to 8:5 (Al:V), and it also indicates that the alloy compounds in this region were mostly $A1_8V_5$. The amount of flaky $A1_8V_5$ particles decreased with the increase of the electrode heating time, and the holes and cracks gradually disappeared in the product, which is consistent with the theoretical views reported by Liu [32,33]. Therefore, the uniform AlV55 alloy could easily be obtained when the electrode heating time was extended to 9 min in the electrode heating process.

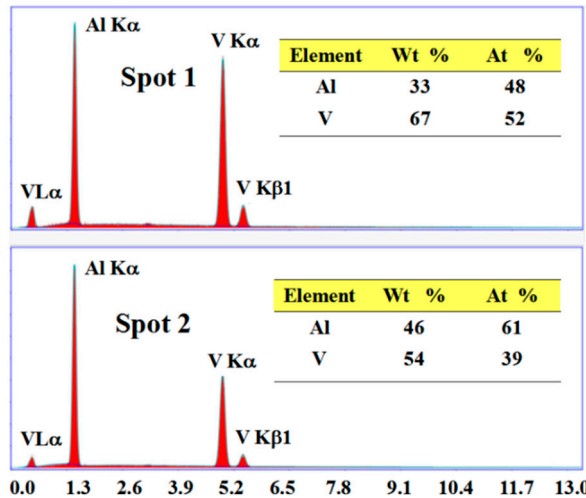

**Figure 10.** Energy dispersive spectrometer (EDS) analysis of the elements distribution in different regions.

## 4. Conclusions

The electrode heating was adopted in this experiment to assist in reduction, and the segregation of the AlV55 alloy was reduced. SEM images analyzed that the cracks in the alloy sample completely disappeared when the mass ratio of $Al/V_2O_5$ was higher than 0.94:1 and electrode heating time was more than 8 min. In the electrode heating process, the analysis results show that Si (0.13 wt%), Fe (0.22 wt%), N (0.007 wt%), C (0.078 wt%), and O (0.051 wt%) impurities in the AlV55 products were reduced, which met the commercial standard (YS/T 579-2014). EDS analyses show that the AlV55 alloy was a non-stoichiometric compound with $A1_8V_5$ and $AlV_3$, and V content ranged from 57.5 to 58.5 wt%. In the electrode heating process, the flaky $A1_8V_5$ particles were decreased, and the uniform AlV55 alloy was obtained when the electrode heating time was controlled at 9 min.

**Author Contributions:** Conceptualization, B.X.; Methodology, H.W. and B.X.; Formal Analysis, L.L. and D.L.; Investigation, H.W., D.L. and B.Y.; Data Curation, H.W., H.W. and B.X.; Writing—Original Draft Preparation, H.W.; Writing—Review & Editing, B.X.; Supervision, L.L. and B.Y.; Project Administration, Y.D.; Funding Acquisition, B.Y.

**Funding:** This research was funded by "National Science Foundation of China" grant number [51734006]; "Science and Technological Talent Cultivation Plan of Yunnan Province, China" grant number [2017HB009]; "Cultivating Plan Program for the Leader in Science and Technology of Yunnan Province" grant number [2014HA003]; "Program for Nonferrous Metals Vacuum Metallurgy Innovation Team of Ministry of Science and Technology" grant number [2014RA4018].

**Acknowledgments:** This work has been founded by National Science Foundation of China (No. 51734006), Science and Technological Talent Cultivation Plan of Yunnan Province, China (2017HB009), the Cultivating Plan Program for the Leader in Science and Technology of Yunnan Province under Grant No. 2014HA003 and The Program for Nonferrous Metals Vacuum Metallurgy Innovation Team of Ministry of Science and Technology under Grant No. 2014RA4018.

**Conflicts of Interest:** The authors declare no conflict of interest. The funders had no role in the design of the study; in the collection, analyses, or interpretation of data; in the writing of the manuscript, and in the decision to publish the results.

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
