# Peer review of "A Novel Method of Fabricating Al-V Intermetallic Alloy through Electrode Heating"

_metals, doi:10.3390/met9050558_

Round 1
Reviewer 1 Report
This paper presents results on efforts to evaluate a novel method for fabricating Al-V intermetallic alloys. The novelty appears to be the use of electrode heating. The writing style of the paper is in need of significant improvements. For example, right at the beginning in the title, in line 3, the word “Electrode” is misspelled (the “c” is missing.) The abstract that follows is not very informative and lacks a complete description of the results that were obtained.
The experimental details that are provided are very limited and do not adequately describe the experiments. More details should be provided on the characterization techniques used. However, the most serious omission is that the scale of the experiments is not known. In Fig. 1 we see a schematic diagram, but there is no information provided on the scale, either size or mass, of the process used. Similarly in Fig. 6, no scale is provided on the assemblage shown. This also raises the question of whether the microscopy images shown in figures 5 and 9 are truly representative of the samples obtained.
However, the most serious limitation of the work is that there is no thermal analysis done, either by simulation or experiment. What temperatures are achieved in the electrode heating process, and what is the temperature distribution? How does it vary over the heating time, and what is the cooling rate? Without this information, there is not a good scientific foundation for the work that is presented here.
Author Response
Dear Reviewer,
Thank you for your nice comments on our article and giving us the opportunity to revise this manuscript. It is with excitement that I resubmit to you a revised version of manuscript “A Novel Method of Fabricating Al-V Intermetallic Alloy through Electrode Heating( Metal-489315 )” for the Metal. Based on your suggestions and comments, we tried our best to improve the manuscript and made some changes in the manuscript. The corresponding descriptions were added by red text in the revised manuscript. Our detailed point-by-point responses to the constructive comments are listed below:
(1) This paper presents results on efforts to evaluate a novel method for fabricating Al-V intermetallic alloys. The novelty appears to be the use of electrode heating. The writing style of the paper is in need of significant improvements. For example, right at the beginning in the title, in line 3, the word “Electrode” is misspelled (the “c” is missing.) The abstract that follows is not very informative and lacks a complete description of the results that were obtained.
Response: Thanks for the kind reminding, we were really sorry for our careless mistakes. We have carefully checked the manuscript and corrected the errors accordingly. Meanwhile, the abstract has been revised and improved. The corresponding descriptions were added by red text in the revised manuscript. The abstract are listed as follows:
A novel method is developed to produce AlV55 alloy through reducing impurities content and component segregation with electrode assisted heating technology. This new process integrates synergistically a few low-cost process techniques including granulation, mixing, and electro-heating to produce AlV55 alloy. During the heating process, the CaO is used as additive in the raw materials. The uniform of AlV55 alloy composition and low impurities content are effectively controlled by this process. The analysis results show that the Si (0.13 wt %), Fe (0.22 wt %), N (0.007 wt %), C (0.078 wt %) and O (0.051 wt %) impurities in the AlV55 products are reduced which meet the commercial standard (TS/T 579-2014), and V content ranging is from 57.5 to 58.5 wt %. when the Al/V2O5 mass ratio is 0.94:1. This method can realize the controllability of the reaction process and is suitable for large-scale industrial production.
(2) The experimental details that are provided are very limited and do not adequately describe the experiments. More details should be provided on the characterization techniques used. However, the most serious omission is that the scale of the experiments is not known. In Fig. 1 we see a schematic diagram, but there is no information provided on the scale, either size or mass, of the process used. Similarly in Fig. 6, no scale is provided on the assemblage shown. This also raises the question of whether the microscopy images shown in figures 5 and 9 are truly representative of the samples obtained.
Response: Thank you for your kind suggestion. As suggested by the reviewer, we have added more details to explain the furnace of temperature and voltage parameters.
The parameters of graphite electrode are as follows: length: L=160cm, diameter: D=25cm, resistivity is 6.5-7.0mΩ·m, the density of electrode is 1.6-1.7g/cm3, and carbon mass fraction greater than 99.5%. The inner diameter of the bottom of the furnace is 10(D1), the top is 11250px(D2), the part below the slag surface is made of Al2O3, the upper part is made of slag(Al2O3·CaO), and the thickness is 750px. Fig. 5 and Fig. 9 are samples obtained under different experimental conditions.
(3) However, the most serious limitation of the work is that there is no thermal analysis done, either by simulation or experiment. What temperatures are achieved in the electrode heating process, and what is the temperature distribution? How does it vary over the heating time, and what is the cooling rate? Without this information, there is not a good scientific foundation for the work that is presented here.
Response: Thank you for your valuable suggestions. The purpose of electrode heating is to provide better heat for the alloy and make the alloy composition more uniform. At present, the temperature simulation work is carried out, which mainly simulates the distribution of temperature field and the change of cooling rate. The temperature of measurement results show that show that the material is heated by the electrode heating which temperature range of alloy is 1973K to 2173K. It is mainly related with the heating time. The temperature of the alloy is not significantly change when the heating time more than 10min.
Sincerely
Baoqiang Xu

Reviewer 2 Report
The manuscript is devoted to the development of a new method for producing an Al-V intermetallic alloy using a heated electrode. The work was done on a topical subject, however, it requires some corrections, in particular:
line 3: typo in the word "Electrode".
line 19: please specify number of the standard.
line 22: put the keyword "Al-V alloy" in the first place; replace the keyword "reduction" with "aluminothermic reduction".
line 66: the sizes of V2O5, Al granules and the size of CaO powder are not given.
line 69: Al2O3 is indicated as a raw material, but it is not described what it is used for.
line 69, Table 1: The purity of the raw materials in the state of supply is indicated, however, to assess the influence of various factors on the contamination of the final product with impurities, it is also advisable to provide data on the factual chemical composition.
line 78: mixing modes in a blender is not specified.
line 82, 83: Are the refractory characteristics of the obtained slag composition sufficient for its use as a lining? (see for example 10.1016/S0272-8842(00)00037-7) According to the state diagram, relatively low-melting compounds can also be formed in the CaO-Al2O3 system. What is the motivation for this application of slag?
line 116: What is a "lager crystal particle"?
line 120: apparently, it should not be about "temperature improvement", but about a more uniform distribution of temperature throughout the ingot volume, about a decrease in temperature gradients.
line 126: It is necessary to present the Gibbs energy calculation method with the sources of the thermodynamic data used (in Section 2).
line 141: Regarding the melting point of 1663 K, what is the composition (ratio of components)?
line 158: Table 2 is better to place in section 2
line 204, Fig. 8: On the ordinate axis there are negative values. Can the content of vanadium in the slag be less than zero?
line 212, Fig. 9: The data on the local elemental composition of the ingot are given. But how controlled the uniformity of the composition and contamination by impurities in the macrovolume ingot? What is the statistics, sampling error, etc.?
line 241: Fill out the section Author Contributions.
line 248: Fill out the section Conflicts of Interest.
The text of the manuscript as a whole contains numerous spelling errors and needs to be completely revised by proficient English speaker.
Author Response
Dear Reviewer,
Thank you for your nice comments on our article and giving us the opportunity to revise this manuscript. It is with excitement that I resubmit to you a revised version of manuscript “A Novel Method of Fabricating Al-V Intermetallic Alloy through Electrode Heating( Metal-489315 )” for the Metal. Based on your suggestions and comments, we tried our best to improve the manuscript and made some changes in the manuscript. The corresponding descriptions were added by red text in the revised manuscript. Our detailed point-by-point responses to the constructive comments are listed below:
(1) line 3: typo in the word "Electrode".
Response: Thank you for your careful work, we have changed the word "Electrode".
(2) line 19: please specify number of the standard.
Response: Thank you again for your suggestion, we have added the standard (TS/T 579-2014).
(3) line 22: put the keyword "Al-V alloy" in the first place; replace the keyword "reduction" with "aluminothermic reduction".
Response: Thank you for your careful work. As the reviewer said, we have changed the keyword.
(4) line 66: the sizes of V2O5, Al granules and the size of CaO powder are not given.
Response: We sincerely appreciate the valuable advice. The size of raw material particles are added in Table 1.
(5) line 69: Al2O3 is indicated as a raw material, but it is not described what it is used for. line 69, Table 1: The purity of the raw materials in the state of supply is indicated, however, to assess the influence of various factors on the contamination of the final product with impurities, it is also advisable to provide data on the factual chemical composition.
Response: Thank you for your suggestion. Al2O3 is mainly used for lining of furnace, and the slag(CaO·Al2O3) is used as lining material in the top part of furnace body where it may not contact with materials. The materials content has been supplemented, as shown in the Table 1.
(6) line 78: mixing modes in a blender is not specified.
Response: We sincerely appreciate the valuable advice, the mixing mode is as follows: The mechanical blender was used in the experiment, which speed is 5r/min.
(7) line 82, 83: Are the refractory characteristics of the obtained slag composition sufficient for its use as a lining? (see for example 10.1016/S0272-8842(00)00037-7) According to the state diagram, relatively low-melting compounds can also be formed in the CaO-Al2O3 system. What is the motivation for this application of slag?
Response: Thank you for your careful work. The main reasons for using slag as refractory material on the top of furnace lining are as follows : (1) solid waste slag is recycled; (2) as the furnace lining in the low temperature region, the slag is the by-product of the reaction which can avoid the impurities bring into the alloy quality.
(8) line 116: What is a "lager crystal particle"?
Response: Thank you for your careful work. This is our carelessness,it is " another crystal particle"
(9) line 120: apparently, it should not be about "temperature improvement", but about a more uniform distribution of temperature throughout the ingot volume, about a decrease in temperature gradients.
Response: Thank you again for your suggestion, we have modified this incorrect expression. "temperature improvement" changed into "temperature gradient improvement" .
(10) line 126: It is necessary to present the Gibbs energy calculation method with the sources of the thermodynamic data used (in Section 2).
Response: Thank you for your advice, we have added the details of thermodynamic calculations to the Section 2.
(11) line 141: Regarding the melting point of 1663 K, what is the composition (ratio of components)?
Response: Thank you for your careful work. The CaO in the slag is mainly in the form of CaO·2Al2O3 which contribute to the melting point of the slag.
(12) line 158: Table 2 is better to place in section 2
Response: Thank you for your suggestion, we have removed the Table 2 to in section 2.
(13) line 204, Fig. 8: On the ordinate axis there are negative values. Can the content of vanadium in the slag be less than zero?
Response: Thank you for your careful work. As the reviewer said, we have changed the Fig.8.
(14) line 212, Fig. 9: The data on the local elemental composition of the ingot are given. But how controlled the uniformity of the composition and contamination by impurities in the macrovolume ingot? What is the statistics, sampling error, etc.?
Response: Thank you for your careful work. The samples in Fig. 9 are representative of common products, and the differences between samples are caused by the different electrode heating time. Therefore, the longer the raw materials is heated, the more effectively the temperature gradient in the alloy is controlled, so the composition and impurity content (O,N,C) of the alloy are improved. We have conducted a large number of experiments, but there may be some errors in strict statistics. We will improve and verify the experimental results in the future.
(15) line 241: Fill out the section Author Contributions.
Response: Thank you for your reminding. The manuscript is completed with the participation and efforts of the co-authors, and the authors’ contributions as follows:
Conceptualization, B.X.; Methodology, H.W. and B.X.; Formal Analysis, L.L. and D.L.; Investigation, H.W., D.L. and B.Y.; Data Curation, H.W., H.W. and B.X.; Writing-Original Draft Preparation, H.W.; Writing-Review & Editing, B.X.; Supervision, L.L. and B.Y.; Project Administration, Y.D.; Funding Acquisition, B.Y.
(17)line 248: Fill out the section Conflicts of Interest.
Response: Thank you for your reminding. The authors declare no conflict of interest. The funders had no role in the design of the study; in the collection, analyses, or interpretation of data; in the writing of the manuscript, and in the decision to publish the results.
Sincerely
Baoqiang Xu

Reviewer 3 Report
The article title describes quite well the work content, and its focus has important practical applications to product master alloys.
There is a number of questions in the course of the presentation and discussion of the results obtained by the authors:
Why authors don’t use established terms when describe the results and bring the names. For example, in the sentence «Vanadium and aluminum vanadium (AlV) alloys are used the term…» «additives», instead of «Master Alloy», although there are references to works numbered [12, and 31]. The authors also use the term «self-propagating reaction», although in the world literature this name has quite common forms, namely, «Combustion Synthesis, Self-propagating high temperature synthesis, SHS-metallurgy».
In the section «Results and Discussion» the authors present a reaction equation describing the change of enthalpy, although a change of Gibbs energy should be given. In addition, when writes the reaction equation, the authors argue that the masses of the products are equal to the masses of the initial reagents. In reality, the process proceeds quite differently, in the region of the aluminum melting point (the authors indicate 943 K), the reaction of vanadium pentoxide reduction with aluminum starts, at the same temperature vanadium pentoxide melts, the process is characterized by significant energy release with increasing temperature to 2000 K and higher, and at this temperature boils and decomposes vanadium pentoxide. Boiling leads to a variation in both the starting materials and the products. Thus, the mass of the reaction products cannot be equal to the mass of the initial reagents. By this reason it should be added the reaction to decompose vanadium pentoxide into a chain of chemical reactions, an oxygen evolution reaction.
The article contains a number of minor flaws:
1. It is necessary to correct a grammatical error in the article title.
2. Table 1 should give the size of the granules of the initial reagents, since this is a very important parameter that influences the mechanism and kinetics of the process.
3. The authors present the results of the content of non-metallic impurities such as carbon and oxygen, but they don’t give the data in what form they exist. Oxygen can be either in dissolved form or in bound form, for example, Al2O3. Carbon can form vanadium carbides.
4. The results of EDS analysis (Figure 10) should be given to the nearest tenth of percent.
Author Response
Dear Reviewer,
Thank you for your nice comments on our article and giving us the opportunity to revise this manuscript. It is with excitement that I resubmit to you a revised version of manuscript “A Novel Method of Fabricating Al-V Intermetallic Alloy through Electrode Heating( Metal-489315 )” for the Metal. Based on your suggestions and comments, we tried our best to improve the manuscript and made some changes in the manuscript. The corresponding descriptions were added by red text in the revised manuscript. Our detailed point-by-point responses to the constructive comments are listed below:
(1) Why authors don’t use established terms when describe the results and bring the names. For example, in the sentence «Vanadium and aluminum vanadium (AlV) alloys are used the term…» «additives», instead of «Master Alloy», although there are references to works numbered [12, and 31]. The authors also use the term «self-propagating reaction», although in the world literature this name has quite common forms, namely, «Combustion Synthesis, Self-propagating high temperature synthesis, SHS-metallurgy».
Response: Thank you very much for reminding us, we have used common terms in the this article.
In addition, when writes the reaction equation, the authors argue that the masses of the products are equal to the masses of the initial reagents. In reality, the process proceeds quite differently, in the region of the aluminum melting point (the authors indicate 943 K), the reaction of vanadium pentoxide reduction with aluminum starts, at the same temperature vanadium pentoxide melts, the process is characterized by significant energy release with increasing temperature to 2000 K and higher, and at this temperature boils and decomposes vanadium pentoxide. Boiling leads to a variation in both the starting materials and the products. Thus, the mass of the reaction products cannot be equal to the mass of the initial reagents. By this reason it should be added the reaction to decompose vanadium pentoxide into a chain of chemical reactions, an oxygen evolution reaction.
Response: Thank you for your suggestion. During the experiment, about 5% of the raw materials (V2O5, Al) were volatilized due to high temperature, so a certain amount of CaO and gold alloy powder was added to reduce the effect of boiling on volatilization of raw materials, finally, the volatile amount of raw materials is less than 1%,which can be almost ignored. In the later research, we will further focus on the influence of boiling on the reaction process in combination with simulation.
(1)It is necessary to correct a grammatical error in the article title.
Response: Thank you for your reminding, we have changed the title.
(2)Table 1 should give the size of the granules of the initial reagents, since this is a very important parameter that influences the mechanism and kinetics of the process.
Response: Thank you for your careful work, we have added the details about the materials in the Table 1.
(3)The authors present the results of the content of non-metallic impurities such as carbon and oxygen, but they don’t give the data in what form they exist. Oxygen can be either in dissolved form or in bound form, for example, Al2O3. Carbon can form vanadium carbides.
Response: Thank you for your suggestion. We have described the form of impurities present in the alloy for example, the impurities of N and O form Al2O3 and AlN, however, C forms CV with vanadium.
(4)The results of EDS analysis (Figure 10) should be given to the nearest tenth of percent.
Response: We sincerely appreciate the valuable advice. As your said, we have changed the Fig. 10.
Sincerely
Baoqiang Xu
Round 2
Reviewer 1 Report
The revisions that are presented in the newer version are useful but also raise some additional questions. The size of the furnace is now given, so we see that it is of considerable size. During re-melting and solidification of alloys, it is well-know that solutes and impurities, as well as the microstructure, can vary considerable with position within the ingot. The micrographs are on the 100-micron scale, so the question comes up as to where the samples were taken from within the ingot and how representative they are of the entire ingot. Similarly, with the impurity content shown in Fig. 7, where within the ingot were samples taken from? Is there macroscopic impurity segregation across the ingot? Were representative samples taken from across the width and depth of the ingot? If so, what were the ranges found? The error bars in Fig. 7 are quite small, but what do they represent?
Also of interest is the response to the question about the temperature distribution. The authors state that the “The temperature of measurement results show that show that the material is heated by the electrode heating which temperature range of alloy is more than 1923 K, and it is mainly related with the heating time.” The idea the temperature is related to heating time is troubling, as Figs. 7 and 8 imply that time is the controlling variable and that temperature is constant. If temperature is changing with time in an uncertain manner, the work is not very reproducible, and the scientific value of the results are significantly reduced.
In lines 200/201, they refer to a commercial 200 standard (TS/T 579-2014) but do not state what that is in terms of impurity content. I expect most readers would not be very familiar with that standard, so it should be given here.
The English language use in the paper is still in need of significant work.
For these reasons, I do not believe this paper is yet suitable for publication.
Author Response
Dear Reviewer,
Thank you for your nice comments on our article and giving us the opportunity to revise this manuscript. It is with excitement that I resubmit to you a revised version of manuscript “A Novel Method of Fabricating Al-V Intermetallic Alloy through Electrode Heating( Metal-489315 )” for the Metals. Based on your suggestions and comments, we tried our best to improve the manuscript and made some changes in the manuscript. The corresponding descriptions were added by red text in the revised manuscript. Our detailed point-by-point responses to the constructive comments are listed below:
(1) The revisions that are presented in the newer version are useful but also raise some additional questions. The size of the furnace is now given, so we see that it is of considerable size. During re-melting and solidification of alloys, it is well-know that solutes and impurities, as well as the microstructure, can vary considerable with position within the ingot. The micrographs are on the 100-micron scale, so the question comes up as to where the samples were taken from within the ingot and how representative they are of the entire ingot. Similarly, with the impurity content shown in Fig. 7, where within the ingot were samples taken from? Is there macroscopic impurity segregation across the ingot? Were representative samples taken from across the width and depth of the ingot? If so, what were the ranges found? The error bars in Fig. 7 are quite small, but what do they represent?
Response: Thanks for the kind reminding, we were really sorry for our careless mistakes. We did not give the size of alloy ingot and furnace body in detail, we will give the detailed explanation as follows. The corresponding descriptions were added by red text in the revised manuscript.
The size given in the manuscript is the size of the furnace shell. Because the furnace body has a certain thickness of resistant material(750px),so the diameter size of base the alloy ingot is about 1000px(d2), the diameter at the top of the slag layer is about 2000px(d2), as shown in figure6. In addition, the alloy ingot is broken after surface grinding. The analysis sample comes from different positions, and the analysis result can represent the whole alloy ingot. So the Figure 7 is a representative sample.
(2) Also of interest is the response to the question about the temperature distribution. The authors state that the “The temperature of measurement results show that show that the material is heated by the electrode heating which temperature range of alloy is more than 1923 K, and it is mainly related with the heating time.” The idea the temperature is related to heating time is troubling, as Figs. 7 and 8 imply that time is the controlling variable and that temperature is constant. If temperature is changing with time in an uncertain manner, the work is not very reproducible, and the scientific value of the results are significantly reduced.
Response: Thank you for your kind suggestion. We are very sorry that the main purpose of heating was not clearly defined in the manuscript. The purpose of heating with electrodes is mainly to supplement the heat, which promotes the reduction reaction is carried out thoroughly, and the cooling rate of the alloy is improved. It would avoid segregation and impurity entering caused by rapid cooling of the alloy. The corresponding descriptions were added by red text in the revised manuscript.
(3) In lines 200/201, they refer to a commercial 200 standard (TS/T 579-2014) but do not state what that is in terms of impurity content. I expect most readers would not be very familiar with that standard, so it should be given here.
Response: Thank you for your valuable suggestions, we have listed the requirements for commercial application of AlV55 alloy in Table 2. The corresponding descriptions were added by red text in the revised manuscript.
(4) The English language use in the paper is still in need of significant work.
Response: We appreciate for your warm work earnestly, we feel sorry for our mistake, we have professionally revised the manuscript in English language. The corresponding descriptions were added by red text in the revised manuscript.
Sincerely
Baoqiang Xu

Reviewer 2 Report
The authors have worked well and made a lot of corrections. The article can be recommended for publication, however there are some shortcomings:
line 134: databases of initial thermodynamic data used for calculations are still not specified
line 198: it is better to write VC instead CV
lines 259-260: Supervision, Project Administration and Funding Acquisition (L.L., B.Y., Y.D.) are not a sufficient contribution to recognize the full-fledged co-authors of the paper (see authorship and contributorship criteria of COPE)
Author Response
Dear Reviewer,
Thank you for your nice comments on our article and giving us the opportunity to revise this manuscript. It is with excitement that I resubmit to you a revised version of manuscript “A Novel Method of Fabricating Al-V Intermetallic Alloy through Electrode Heating( Metal-489315 )” for the Metals. Based on your suggestions and comments, we tried our best to improve the manuscript and made some changes in the manuscript again. The corresponding descriptions were added by red text in the revised manuscript. Our detailed point-by-point responses to the constructive comments are listed below:
(1) line 134: databases of initial thermodynamic data used for calculations are still not specified.
Response: Thank you for your careful work, we have provided a source of thermodynamic data and the reference[23].
(2) line 198: it is better to write VC instead CV.
Response: Thank you again for your very meaningful advice, we have changed it in the article.
(3) line 259-260: Supervision, Project Administration and Funding Acquisition (L.L., B.Y., Y.D.) are not a sufficient contribution to recognize the full-fledged co-authors of the paper (see authorship and contributor ship criteria of COPE).
Response: Thank you for your careful work. As the reviewer said, we have changed this part.
Sincerely
Baoqiang Xu
Reviewer 3 Report
Authors write:
@Thank you for your suggestion. During the experiment, about 5% of the raw materials (V2O5, Al) were volatilized due to high temperature, so a certain amount of CaO and gold alloy powder was added to reduce the effect of boiling on volatilization of raw materials, finally, the volatile amount of raw materials is less than 1%,which can be almost ignored. In the later research, we will further focus on the influence of boiling on the reaction process in combination with simulation@
In this connection what has been said, it is necessary to present data on the alloy into Table 1. In the continuation of this thought, the question arises: Which phase does the remnants (traces of impurity) of this alloy get into the slag or master alloy, there is no confirmation by the data of scanning electron microscopy and chemical analysis?
Once again I draw attention to the need to change the presentation of results in the section @ 3.1. Theoretical Analysis @ in terms of Gibbs energy (ΔG), and not through enthalpy (ΔH), formulas 1 – 3.
Author Response
Dear Reviewer,
Thank you for your nice comments on our article and giving us the opportunity to revise this manuscript. It is with excitement that I resubmit to you a revised version of manuscript “A Novel Method of Fabricating Al-V Intermetallic Alloy through Electrode Heating( Metal-489315 )” for the Metals. Based on your suggestions and comments, we tried our best to improve the manuscript and made some changes in the manuscript. Our detailed point-by-point responses to the constructive comments are listed below:
(1) In this connection what has been said, it is necessary to present data on the alloy into Table 1. In the continuation of this thought, the question arises: Which phase does the remnants (traces of impurity) of this alloy get into the slag or master alloy, there is no confirmation by the data of scanning electron microscopy and chemical analysis?
Response: Thank you for your careful work. As the reviewer said, we have added the composition of theAlV55 particles alloy composition is shown in Table 1. As the raw material composition of the alloy used is lower than the commercial standard, so it will not have obvious influence on the alloy product. The experimental results also confirm the feasibility of this technology. The corresponding descriptions were added by red text in the revised manuscript.
(2) Once again I draw attention to the need to change the presentation of results in the section @ 3.1. Theoretical Analysis @ in terms of Gibbs energy (ΔG), and not through enthalpy (ΔH), formulas 1 – 3.
Response: Thank you for your kind suggestion. As suggested by the reviewer, we have change the presentation of results in the section 3.1 about theoretical analysis. The corresponding descriptions were added by red text in the revised manuscript.
Sincerely
Baoqiang Xu
Round 3
Reviewer 1 Report
I have no additional comments beyond what was given in the previous review.
Author Response
Dear Reviewer,
Thank you for your nice comments on our article and giving us the opportunity to revise this manuscript. It is with excitement that I resubmit to you a revised version of manuscript “A Novel Method of Fabricating Al-V Intermetallic Alloy through Electrode Heating( Metal-489315 )” for the Metal. Based on your suggestions and comments, we tried our best to improve the research design and made some changes about methods described in the manuscript. The corresponding descriptions were added by red text in the revised manuscript.
Sincerely
Baoqiang Xu

Reviewer 3 Report
Dear authors!
You have done a great job, congratulations!
In the presented form, the article can be published, but it would be desirable that the roughness and flaws be eliminated.
Signatures of graphic material should be made in the same style through the manuscript; indexes of chemical compounds should be written in the subscript style, for example "Al2O3" replace with "Al2O3".
There are sentences with the presence of roughness in the text of the article , for example
"Theoretically,
line: 196 more heat during the process can help avoid rapid solidification, however, it is known that
line: 197 carbon would diffuse into the alloy if heating too long. The temperature of measurement results
line: 198 show that show that the material is heated by the electrode heating which temperature range of
line: 199 alloy is more than 1923 K(about1980-2000K), and it is mainly related with the heating time."
Author Response
Dear Reviewer,
Thank you for your nice comments on our article and giving us the opportunity to revise this manuscript. It is with excitement that I resubmit to you a revised version of manuscript “A Novel Method of Fabricating Al-V Intermetallic Alloy through Electrode Heating( Metal-489315 )” for the Metals. Based on your suggestions and comments, we tried our best to improve the manuscript and made some changes in the manuscript. Our detailed point-by-point responses to the constructive comments are listed below:
Signatures of graphic material should be made in the same style through the manuscript; indexes of chemical compounds should be written in the subscript style, for example "Al2O3" replace with "Al2O3".
Response: Thank you for your careful work. As the reviewer said, we have changed the indexes of chemical compounds as shown in Figure 6.
(2) There are sentences with the presence of roughness in the text of the article , for example "Theoretically,
Response: Thanks for the kind reminding, we were really sorry for our careless mistakes. We have carefully checked the manuscript and corrected the errors accordingly.
(3) line: 196 more heat during the process can help avoid rapid solidification, however, it is known that
Response: Thank you for your kind suggestion. As suggested by the reviewer, we have revised more details for the sentence.
(4) line: 197 carbon would diffuse into the alloy if heating too long. The temperature of measurement results
line: 198 show that show that the material is heated by the electrode heating which temperature range f
line: 199 alloy is more than 1923 K(about1980-2000K), and it is mainly related with the heating time."
Response: Thank you for your careful work. We have revised this part(line197-199) in the manuscript.
Sincerely
Baoqiang Xu
